# Running-Induced Metabolic and Physiological Responses Using New Zealand Blackcurrant Extract in a Male Ultra-Endurance Runner: A Case Study

**DOI:** 10.3390/jfmk7040104

**Published:** 2022-11-23

**Authors:** Mark E. T. Willems, Andrew R. Briggs

**Affiliations:** Institute of Sport, Nursing and Allied Health, University of Chichester, College Lane, Chichester PO19 6PE, UK

**Keywords:** blackcurrant, anthocyanins, ultra-endurance, running, substrate oxidation, heart rate, core body temperature

## Abstract

Physical training for ultra-endurance running provides physiological adaptations for exercise-induced substrate oxidation. We examined the effects of New Zealand blackcurrant (NZBC) extract on running-induced metabolic and physiological responses in a male amateur ultra-endurance runner (age: 40 years, body mass: 65.9 kg, BMI: 23.1 kg·m^−2^, body fat: 14.7%, $\dot{V}$O_2max_: 55.3 mL·kg^−1^·min^−1^, resting heart rate: 45 beats·min^−1^, running history: 6 years, marathons: 20, ultra-marathons: 28, weekly training distance: ~80 km, weekly running time: ~9 h). Indirect calorimetry was used and heart rate recorded at 15 min intervals during 120 min of treadmill running (speed: 10.5 km·h^−1^, 58% $\dot{V}$O_2max_) in an environmental chamber (temperature: ~26 °C, relative humidity: ~70%) at baseline and following 7 days intake of NZBC extract (210 mg of anthocyanins·day^−1^) with constant monitoring of core temperature. The male runner had unlimited access to water and consumed a 100-kcal energy gel at 40- and 80 min during the 120 min run. There were no differences (mean of 8, 15 min measurements) for minute ventilation, oxygen uptake, carbon dioxide production and core temperature. With NZBC extract, the respiratory exchange ratio was 0.02 units lower, carbohydrate oxidation was 11% lower and fat oxidation was 23% higher (control: 0.39 ± 0.08, NZBC extract: 0.48 ± 0.12 g·min^−1^, *p* < 0.01). Intake of the energy gel did not abolish the enhanced fat oxidation by NZBC extract. Seven days’ intake of New Zealand blackcurrant extract altered exercise-induced substrate oxidation in a male amateur ultra-endurance runner covering a half-marathon distance in 2 h. More studies are required to address whether intake of New Zealand blackcurrant extract provides a nutritional ergogenic effect for ultra-endurance athletes to enhance exercise performance.

## 1. Introduction

Physical training for ultra-endurance events includes repeated sessions of moderate-intensity continuous exercise of long duration [1]. Key training adaptations obtained by responders to regular moderate-intensity continuous exercise are mitochondrial biogenesis [2], an increase in maximal oxygen uptake [3], an increase in skeletal muscle capillarization [4] and enhanced exercise-induced whole-body fat oxidation [5]. In male ironman athletes, maximal fat oxidation and peak oxygen consumption showed significant negative correlations with the ironman race time [6]. The observation by Frandsen et al. [6] indicates the importance for whole-body fat oxidation during ultra-endurance events. Ultra-endurance athletes may benefit from enhanced whole-body fat oxidation during racing events by attenuating the rate of glycogen utilization. 

Dietary intake studies with low-carbohydrate, high-fat diets reported enhanced whole-body fat oxidation in elite race walkers [7]. However, the observations of impaired exercise performance with low-carbohydrate, high-fat diets [7,8] may have dampened the interest of it being a popular nutritional strategy among elite ultra-endurance athletes [9,10]. In ultra-endurance athletes, dietary and training practices are expected to optimize the required physiological and structural adaptations that may minimize or even discount the effectiveness of dietary supplements that may affect substrate oxidation. There is evidence on the impact that training status can have on the response to supplementation. For example, the response to dietary nitrate intake was better in less-trained athletes for enhancing time-trial performance [11]. In addition, in response to caffeine, trained males showed fewer effects for a 3 km cycling time trial [12]. During competitive events, ultra-endurance athletes consume primarily carbohydrate [13,14], and there is evidence for caffeine use [14,15,16]. It is not clear whether the rationale for caffeine intake is to affect exercise-induced substrate oxidation [17]. 

Recently, a number of studies have provided evidence of an increase of exercise-induced fat oxidation by the intake of New Zealand blackcurrant extract in non-elite endurance athletes [18,19,20,21] and recreationally active individuals [22,23]. New Zealand blackcurrant extract is an anthocyanin-rich supplement consisting primarily of the anthocyanins delphinidin-3-rutinoside, delphinidin-3-glucoside, cyanidin-3-rutinoside and cyanidin-3-glucoside. New Zealand blackcurrant extract may enhance lipolysis [20] and, by improving peripheral blood flow [24], increase the delivery of free fatty acids to the working skeletal muscles. It is not known whether ultra-endurance athletes would respond to supplements that are known to enhance exercise-induced fat oxidation in other cohorts.

Due to the logistical challenges of recruitment of ultra-endurance athletes for a laboratory-based study, we adopted a case-study approach [16,25]. The aim of the present study was to examine in a trained male amateur ultra-endurance runner the effects of New Zealand blackcurrant extract on primarily whole-body substrate oxidation during endurance exercise in the time period between two 100 mile running events. In addition, because it is a common nutritional strategy to take carbohydrates during endurance exercise, we allowed the intake of energy gels to examine whether that would blunt the potential exercise-induced fat oxidation by the effects of New Zealand blackcurrant extract.

## 2. Materials and Methods

One male amateur ultra-endurance runner (age: 40 years, height: 169 cm, body mass: 65.9 kg, $\dot{V}$O_2max_: 55.3 mL·kg^−1^·min^−1^) who had entered three ultra-marathon events over the summer of 2021 [i.e., Thames Path 100 miles (~161 km, 8 May 2021, United Kingdom), South Downs Way 100 miles (~161 km, 12 June 2021, United Kingdom), and the 145 km Sur les Traces des Ducs de Savoie (24 August 2021, France)] volunteered to participate in the present study with testing between the May and June event. The participant had been running for 6 years, completed 20 marathons and 28 ultramarathons, with an average weekly running distance of 80 km in an average of 9 h of training per week. Written informed consent was provided after being informed of the experimental procedures, potential risks and right to withdrawal. The participant completed a health history questionnaire and had no conditions or supplement use that could interfere with the metabolic and physiological parameters of the study. The study was approved according to the Research Ethics University Policy (approval code: 1705671) of the University of Chichester (United Kingdom).

### 2.1. Experimental Design

The participant visited the Exercise Physiology laboratory at the University of Chichester on three occasions. Details of the visits are described below. In short, in the first visit, measurements of physiological parameters and hydration status were taken. The participant performed an incremental submaximal running protocol and incremental running protocol to exhaustion to determine maximum oxygen uptake and the percentage of maximum oxygen uptake at the running speed of 10.5 km·h^−1^ for the 2 h treadmill run in visits two (on 22 May 2021, 14 days after completion of a 100-mile running event) and three (on 3 June 2021, 9 days before the next 100-mile running event). In visits two and three, the participant performed the 2 h half-marathon distance treadmill run without and after 7-day intake of New Zealand blackcurrant extract with measurement of physiological and metabolic parameters. All visits for the study were initiated in the late afternoon or early evening, i.e., 18.00 ± 1.00 h. The participant abstained from strenuous exercise and alcohol for 48 hrs prior to each laboratory visit but was allowed to continue with his habitual exercise program. Dietary intake was recorded for 24 hrs prior to visit two and the participant was instructed to replicate this diet for 24 hrs before visit three. 

### 2.2. Visit One—Preliminary Measurements

Following measurements of body mass (Seca Model 876, Seca Ltd., Birmingham, UK) and stature (Holtain Stadiometer, Crymych, Dyfed, UK), blood samples were taken with the finger prick method for resting haemoglobin (13.1 mg·dL^−1^), haematocrit (40%), glucose (5.57 mmol·L^−1^) and lactate (1.26 mmol·L^−1^) (YSI 2300 STAT PLUS, Analytical Technologies, Farnborough, Hants, UK) and resting heart rate recorded (45 beats·min^−1^). Subsequently, the participant completed an incremental submaximal test and incremental test to volitional exhaustion (see below). 

#### Incremental Submaximal Test and Incremental Test to Exhaustion

For the incremental submaximal test, the participant completed an eight-stage exercise test on a motorized treadmill (Woodway ELG70, Cranlea & Co, Birmingham, UK) with the gradient set at 1% incline [26], a starting speed of 8 km∙h^−1^ with increments of 0.75 km∙h^−1^, and each stage lasting 4 min. During the final 90 s of each stage, expired air was collected in Douglas bags for analysis. This test and the oxygen uptake quantification allows calculation of the % of maximum oxygen uptake for a running speed of 10.5 km∙h^−1^ for visits two and three. Following an active recovery for 15 min after the eight-stage exercise test, an incremental test with a starting running speed of 12 km∙h^−1^ and increments of 0.1 km∙h^−1^ every 6 s until volitional exhaustion was performed for determination of maximum oxygen uptake. Expired air was collected in the final 4 min of the test with Douglas bags. Heart rate was measured using short range telemetry (RS400, Polar Electro UK Ltd., Warwick, UK). Maximum heart rate was 178 beats∙min^−1^. For the measurements of respiratory parameters, expired air was analysed for fractions of oxygen and carbon dioxide using a calibrated paramagnetic oxygen analyser and an infrared carbon dioxide analyser (series 1440; Servomex plc, Crowborough, UK). The volume of expired air was measured with a calibrated dry gas meter (Harvard Apparatus Ltd., Edenbridge, UK) with simultaneous measurement of expired air temperature during Douglas bag evacuation for gas volume temperature corrections. Barometric pressure was measured using a mercury barometer at the start of each testing session.

### 2.3. Supplementation Strategy for Visit Three

Due to the ~4-week time period between the ultramarathons in May and June to carry out all the testing, and allowing recovery from the May event and rest for the June event, it was decided to supplement only in visit three. In addition, it needs to be noted that visits two and three did not have performance measurements that could have been affected by experimental bias. Seven days before visit three, the participant ingested 600 mg of New Zealand blackcurrant (CurraNZ™, Health Currancy Ltd., Surrey, UK) in two capsules per day, with each containing 105 mg of anthocyanins at breakfast. Company information indicated that the anthocyanin composition of each capsule was 35–50% delphinidin-3-rutinoside, 5–20% delphinidin-3-glucoside, 30–45% cyanidin-3-rutinoside and 3–10% cyanidin-3-glucoside 3–10%. 

### 2.4. Visit Two and Three (with New Zealand Blackcurratn Extract)

Upon arrival for the visits for the 2 h treadmill run, hydration status was confirmed with urine osmolality < 600 mOsm·kg^−1^ [27], followed by a 10 min rest. The participant rested for an additional 10 min on entering the environmental chamber (TISS Model 201003-1, TIS Services UK, Medstead, Hampshire, UK). Environmental conditions for ambient temperature and humidity were 26 °C and 68.7% (Kestrel Meter 5400 Heat Stress Tracker, Kestrel Meters, Boothwyn, PA, USA) for visit two and 25.8 °C and 70.4% for visit three, calculated with 15 min recordings during the 2 h run.

The treadmill was set to 1% gradient [26] and 10.5 km·h^−1^ for the 2 h run, completing a half marathon distance in speed for 2 h at an intensity of 58% $\dot{V}$O_2max._ Heart rate measurement, expired air collections with Douglas bags and fractions of oxygen and carbon dioxide in the environmental chamber [28] were taken every 15 min during the 2 hr run. During the 2 h run in visits two and three, the participant was allowed to drink water at self-selected times and consumed one 100 kcal energy gel (total carbohydrate: 22 g with total sugar: 7, total fat: 0.5 g, sodium: 60 mg, amino acids: 450 mg) (GU Energy UK) at 40 and 80 min. 

### 2.5. Data Calculations and Statistical Analysis 

The rates of exercise-induced whole-body fat and carbohydrate oxidation during the 2 h treadmill run were calculated with the proposed equations for moderate-to-high intensity (50–75% $\dot{V}$O_2max_) from Jeukendrup and Wallis [29]. Normality was checked with a D’Agostino and Pearson omnibus normality test (GraphPad Prism v5 for Windows). A paired two-tailed *t*-test was used for analysis for the 15 min (i.e., 8 time points) measurements during the 2 h treadmill run in the control and New Zealand blackcurrant extract conditions. Data are reported as mean ± SD and 95% confidence intervals, calculated from the 15 min (i.e., 8 time points) recordings during the 2 h treadmill run. Significance was accepted at *p* < 0.05.

## 3. Results

During the 2 h treadmill run at ~26 °C and ~70% humidity, the participant consumed ad libitum 518 and 464 mL of water in the control and New Zealand blackcurrant (NZBC) extract conditions.

### 3.1. Core temperature

NZBC extract had no effect on core temperature during the 2 h half-marathon treadmill run (control: 38.2 ± 0.2 °C, 95% CI [38.1, 38.4 °C]; NZBC extract 38.2 ± 0.3 °C, 95% CI [38.0, 38.5 °C]; *p* = 0.79).

### 3.2. Physiological Responses

In the NZBC extract condition, there was a higher heart rate of 7 beats·min^−1^ during the 2 h half-marathon treadmill run (control: 142 ± 8 beats·min^−1^, 95% CI [135, 149 beats·min^−1^]; NZBC extract 149 ± 10 beats·min^−1^, 95% CI [140, 157 beats·min^−1^]; *p* < 0.01). NZBC extract had no effect on minute ventilation (control: 48.9 ± 3.5 L·min^−1^, 95% CI [46.0, 51.8 L·min^−1^], NZBC extract: 49.5 ± 3.0 L·min^−1^, 95% CI [47.0, 51.9 L·min^−1^]; *p* = 0.45), oxygen uptake (control: 2.37 ± 0.13 L·min^−1^, 95% CI [2.26, 2.48 L·min^−1^], NZBC extract: 2.38 ± 0.11 L·min^−1^, 95% CI [2.28, 2.46 L·min^−1^]; *p* = 0.45), carbon dioxide production (control: 2.13 ± 0.09 L·min^−1^, 95% CI [2.05, 2.21 L·min^−1^], NZBC extract: 2.08 ± 0.07 L·min^−1^, 95% CI [2.02, 2.13 L·min^−1^]; *p* = 0.12).

### 3.3. Metabolic Responses

NZBC extract lowered the respiratory exchange ratio by 0.02 units during the 2 h treadmill (control: 0.90 ± 0.01, 95% CI [0.89, 0.91]; NZBC extract: 0.88 ± 0.03, 95% CI [0.86, 0.90], *p* < 0.01) (Figure 1a). In addition, NZBC extract provided lower carbohydrate oxidation (control: 1.94 ± 0.12 g·min^−1^, 95% CI [1.85, 2.04 g·min^−1^]; NZBC extract: 1.73 ± 0.21 g·min^−1^, 95% CI [1.55, 1.91 g·min^−1^]; *p* = 0.01) (Figure 1b). NZBC extract provided enhanced exercise-induced fat oxidation by 23% (control: 0.39 ± 0.08 g·min^−1^, 95% CI [0.32, 0.47 g·min^−1^]; NZBC extract (0.48 ± 0.12 g·min^−1^, 95% CI [0.38, 0.59 g·min^−1^]; *p* < 0.01) (Figure 1c). Intake of the 100 kcal energy gels at 40 and 80 min during the 2 h treadmill did not abolish exercise-induced fat oxidation (Figure 1d). NZBC extract was able to affect the mechanisms that regulate substrate oxidation in a male ultra-endurance athlete. 

## 4. Discussion

The present study provided novel observations on the metabolic effects of 7-day intake of an anthocyanin-rich extract made from New Zealand blackcurrant to alter exercise-induced substrate oxidation in a male ultra-endurance athlete. During a 2 h run, in ~26 °C and 70% relative humidity and covering a half-marathon distance, carbohydrate oxidation was decreased by 11% and fat oxidation was increased by 23%. In addition, fat oxidation was not abolished with intake of 100 kcal energy gels at 40 and 80 min into the run, and the oxygen cost for the 2 h run was not affected. Previous studies in male cohorts have also shown enhanced exercise-induced fat oxidation with 7-day intake of NZBC extract in endurance-trained cyclists during bouts of 10 min cycling at 65% $\dot{V}$O_2max_ by 27% [18] and recreationally active males during treadmill walking at an intensity of five metabolic equivalents by 11% [30]. In addition, in recreationally active males and females in a fasted state, enhanced fat oxidation by 30% was observed during 60 min at 65%$\dot{\;V}$O_2max_ in hot ambient conditions (34 °C and 40% relative humidity) [21]. In the studies of Cook et al. [18], Şahin et al. [30] and Hiles et al. [21], not every participant responded with enhanced exercise-induced fat oxidation with NZBC extract. The male ultra-endurance athlete in the present study can be considered a clear responder to the NZBC extract to alter exercise-induced metabolic responses. Responsiveness to the metabolic effects of intake of NZBC extract seems to be independent of the endurance training status of individuals, assuming that our observations of the male ultra-endurance athlete are representative for a majority of ultra-endurance athletes. Interestingly, the participant was tested very late afternoon/early evening, a time of day in which a previous study in endurance-trained cyclists did not observe enhanced exercise-induced fat oxidation [31].

The ultra-endurance athlete in the present study has probably enhanced integrative biological adaptations by physical training compared to the participants in Cook et al. [18] and Şahin et al. [30]. Ultra-endurance athletes adopt physical training programs, including moderate-intensity continuous training, as a key training modality that adheres to overload and progression training principles [32]. Moderate-intensity continuous training provides numerous cardiovascular, physiological, neuromuscular, and metabolic adaptations [33,34,35]. It is also likely that ultra-endurance-trained individuals have an adapted composition of gut microbiota by training and dietary practices [36] that may affect the conversion of anthocyanins to active metabolites [37]. One of the classic metabolic adaptions from physical training in endurance-trained individuals is the ability to enhance exercise-induced fat oxidation [5]. We can assume that the male ultra-endurance athlete had the adaptations that would contribute to enhanced exercise-induced fat oxidation. The present study therefore indicates that obtained physical training adaptations and mechanisms with respect to energy metabolism for ultra-endurance events are still receptive to a known enhancer of exercise-induced fat oxidation with just 7 days’ intake of an anthocyanin-rich NZBC extract. The aim for endurance athletes to enhance exercise-induced fat oxidation with a dietary intervention (not just a supplement) has also been the focus of many studies. 

Studies with a low carbohydrate/high fat diet have aimed to enhance exercise-induced fat oxidation. Havemann et al. [38] is, as far as we know, the only study that reported on a high-fat diet of less than a week with enhanced exercise-induced fat oxidation and changes in mean RER by 0.06 after 7 days during 60 min cycling at 70% $\dot{V}$O_2peak_. Interestingly, Havemann et al. [38] also reported a higher heart rate during exercise as we observed as well for the participant in the present study. The higher heart rate in the present study is unlikely to be explained by lower glycogen stores [39]. In a study with a dietary intervention over 6 weeks, Prins et al. [40] observed that the low carbohydrate/high fat diet enhanced peak fat oxidation by 25% in competitive recreational distance runners ($\dot{V}$O_2max_: 61.6 ± 3.1 mL·kg^−1^·min^−1^). Although the 25% was a mean for the cohort, it is interesting that our participant had enhanced fat oxidation by 23%. In addition, Che et al. [41] observed a change of 28% in maximal fat oxidation in well-trained runners, including elite athletes (from 0.36 ± 0.12 to 0.46 ± 0.06 g·min^−1^) after a fat adaptation with carbohydrate restoration diet. Therefore, it seems that enhanced exercise-induced fat oxidation obtained with some dietary intervention over weeks in endurance-trained cohorts may be obtained as well with intake of an anthocyanin-rich NZBC extract. The mechanisms for enhanced exercise-induced fat oxidation by NZBC extract is not known, but may involve the availability of free fatty acids [20], the transport to the working tissue by enhanced blood flow [24] with enhanced uptake into the mitochondria, maybe by enhanced activity of carnitine palmitoyltransferase I. The use of free fatty acids by exercising skeletal muscle is the consequence of complex regulation with multiple steps involved, and future studies should focus on the mechanisms of enhanced exercise-induced fat oxidation by NZBC extract. It is common for ultra-endurance athletes to have a nutritional dosing strategy with intake of carbohydrates and caffeine [13], but it is not known whether caffeine is taken for a potential enhanced fat oxidation effect. Future studies are needed to examine whether the intake of NZBC extract is beneficial as part of the nutritional strategy during ultra-endurance events.

### Limitations

We have no information on the cardiovascular, physiological and metabolic adaptations that resulted from the physical training undertaken by the male ultra-endurance athlete to be able to compete in ultra-endurance events when he was participating in the present study. The measurements for the control and NZBC extract condition were taken 14 and 25 days after completion of a 100-mile running event, and we do not know whether the athlete was completely recovered. In addition, it is not known how much recovery time is needed to normalize substrate oxidation following a 100-mile running event.

## 5. Conclusions

Intake of an anthocyanin-rich supplement from New Zealand blackcurrant by a male ultra-endurance athlete was able to alter exercise-induced fat oxidation. Although this was observed in a case study, the observation indicates that New Zealand blackcurrant extract can change exercise-induced substrate oxidation in individuals whose physical training resulted in adaptations of the mechanisms that enhance exercise-induced fat oxidation. Future work is needed to examine the effect of New Zealand blackcurrant in a larger cohort of ultra-endurance athletes, including males and females.

## Figures and Tables

**Figure 1 jfmk-07-00104-f001:**
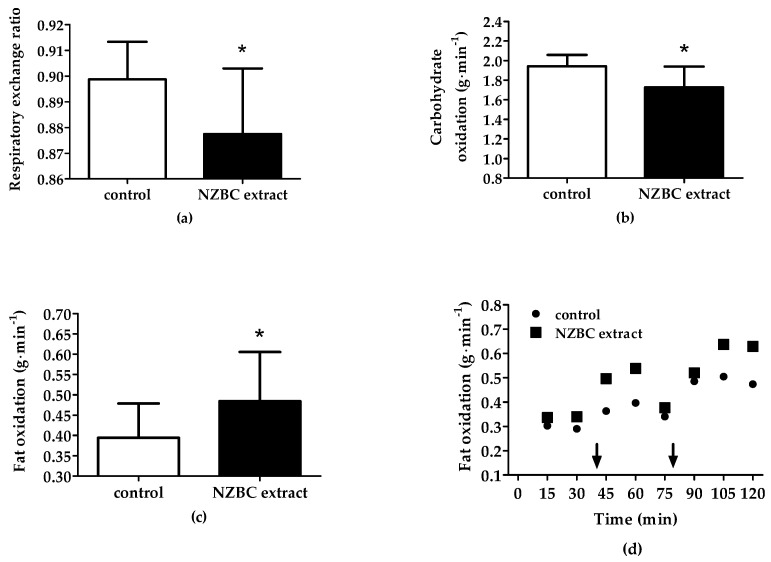
Respiratory exchange ratio (**a**), carbohydrate oxidation (**b**) and fat oxidation (**c**) during the 2 h treadmill run and fat oxidation at 15 min time points during the 2 h treadmill run (**d**). Data in (**a**–**c**) reported as mean ± SD from the 15 min time point recordings. Arrows in (**d**) indicate intake of the 100 kcal energy gels. NZBC, New Zealand blackcurrant, * indicates a difference with control (*p* < 0.05).

## Data Availability

Data will be provided on reasonable request.

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
