# Peer review of "Running-Induced Metabolic and Physiological Responses Using New Zealand Blackcurrant Extract in a Male Ultra-Endurance Runner: A Case Study"

_jfmk, 2022, doi:10.3390/jfmk7040104_

Round 1

Reviewer 1 Report

Thank you for the novel observations on the metabolic effects of NZBC extract in a male ultra-endurance athlete.

I have a question about the impact of test timimg.

Visit 2 is 16 days after 100-mile running event and visit 3 is 27 days after 100-mile running event.

Why was the assessment conducted at a time when the question of recovery from previous marathon (May 22 2021) experience could be raised?

Author Response

We thank the reviewer for the comments. Below our reply to the comment

Thank you for the novel observations on the metabolic effects of NZBC extract in a male ultra-endurance athlete.

I have a question about the impact of test timimg.

Visit 2 is 16 days after 100-mile running event and visit 3 is 27 days after 100-mile running event.

Why was the assessment conducted at a time when the question of recovery from previous marathon (May 22 2021) experience could be raised?

Reply. This is a valid point to raise. However, there is no information in the literature for a 2-week time effect on substrate oxidation during the recovery from an ultra-endurance event. Most studies with the marathon focus on interventions in the first few days after the marathon.

We did mention already as a limitation that measurements for the control and NZBC extract condition were taken at least two weeks after completion of a 100-mile running event and we do not know whether the athlete was complete recovered.

We have added “In addition, it is not known how much recovery time was needed to normalize substrate oxidation following a 100-mile running event”

Reviewer 2 Report

Well written article

I think this can serve as a case study to hopefully inspire future research on this topic

Only one comment/question:

L33: is it moderate or vigorous.  I don’t know the sport all that well, but moderate to me suggests brisk walking vs. vigorous which would be fast walking or running

Author Response

We thank the reviewer for the comments. Below our reply to the comment

Well written article

I think this can serve as a case study to hopefully inspire future research on this topic

Only one comment/question:

L33: is it moderate or vigorous.  I don’t know the sport all that well, but moderate to me suggests brisk walking vs. vigorous which would be fast walking or running

Reply. The athlete completed a half-marathon distance in 2 hours, thus running at 10.5 km/h and at an intensity of 58%O2max. The speed of 10.5 km/h is too fast for brisk/fast walking. The intensity of 58%O2max is moderate intensity as we also used the moderate to high intensity equations to calculate substrate oxidation (i.e. 50-75%O2max) from Jeukendrup and Wallis (2005). That information is all present in the manuscript.